# Cervical cancer screening in a population of black South African women with high HIV prevalence: A cross-sectional study

**Joyce Sikwese Musonda**[1], **Pumla Pamella Sodo**[2]*, **Olalekan Ayo-Yusuf**[3], **Elizabeth Reji**[4], **John Musonda**[2], **Langalibalele Honey Mabuza**[5], **John Velaphi Ndimande**[5], **Jimmy Akii**[2], **Olufemi Babatunde Omole**[2]

1 Department of Family Medicine & Primary Care, University of Pretoria, Pretoria, South Africa, 2 Division of Family Medicine, Department of Family Medicine and Primary Care, University of the Witwatersrand, Johannesburg, South Africa, 3 School of Health Systems and Public Health, University of Pretoria, Pretoria, South Africa, 4 Department of Family Medicine, University of Free State, Bloemfontein, South Africa, 5 Department of Family Medicine & Primary Health Care, Sefako Makgatho Health Sciences University, Pretoria, South Africa

* pumla.sodo@wits.ac.za

**Data Availability Statement:** Our ethics agreement with participants was that the data would only be accessible to the study team due to the sensitive

## Abstract

Cervical cancer is largely preventable through early detection, but screening uptake remains low among black women in South Africa. The purpose of this study was to determine the prevalence and factors associated with cervical cancer screening in the past 10 years among black African women in primary health care (PHC) clinics, in Gauteng Province, South Africa. This was a cross-sectional study involving 672 consecutively recruited black women at cervical cancer screening programs in PHC clinics between 2017 and 2020. An interviewer-administered questionnaire covered socio-demographics, HIV status, sexual history, cervical cancer risk factors knowledge, and screening behaviours in the past 10 years. The mean age of participants was 38 years. More than half (63%) were aged 30–49 years. Most completed high school education (75%), were unemployed (61%), single (60%), and HIV positive (48%). Only 285 (42.4%) of participants reported screening for cervical cancer in the past 10 years. Of participants that reported receiving information on screening, 27.6% (n = 176) and 13.97% (n = 89) did so from healthcare facilities and community platforms respectively. Participants aged 30 years or more were more likely to report for cervical cancer screening as compared to other categories in the past 10 years. The study found low cervical cancer screening prevalence. This calls for health education campaigns and prevention strategies that would target individual patients' contexts and stages of behavioral change. Such strategies must also consider socio-demographic and clinical correlates of cervical cancer screening and promote better integration into PHC services in South Africa.

information contained in the data; hence, it would compromise the HREC ethical standards to allow the data to be publicly available in a public repository, within the manuscript itself or uploaded as supplementary information. We are happy to share the data, or parts of the data, on a case-by-case basis. Please contact Joel Francis (non-author), for such requests at Joel.francis@wits.ac.za.

**Funding:** This study was co-funded by the National Research Foundation grant 93030 for OAY and the Project SHPC 000 fund of the Department of Family medicine, University of the Witwatersrand for OO. The funders had no role in study design, data collection, and analysis, decision to publish, or preparation of the manuscript.

**Competing interests:** The authors have declared that no competing interests exist.

## Introduction

Cervical cancer is an important global public health problem and ranks fourth among the leading causes of cancer-related deaths among women [1–5]. In 2018, an estimated 311,000 deaths and 570,000 new cases of cervical cancer were reported globally, with 85% of these in developing countries. Sub-Saharan Africa recorded the highest age-standardized incidence rates (ASIR) of >40 per 100 000 in 2018, compared to an ASIR of 7.2 and 26.8 per 100,000 for North and Middle Africa respectively [1].

The main risk factor for cervical cancer is infection with the Human Papilloma Virus (HPV) serotypes 16 & 18 [1, 6, 7]. Other risk factors include HIV and Chlamydia trachomatis infections, early sexual debut, multiple sexual partners, smoking, and the use of oral hormonal contraceptives [8]. The best possible protection against cervical cancer is HPV vaccination, early detection and prompt treatment of pre-cancerous conditions [5]. A well-organized population-based cytological screening programmes is therefore crucial and has been found effective in reducing both mortality and morbidity in many Europe, Australia, New Zealand, and North America [9–12]. However, morbidity and mortality remain high in many developing countries consequent to low uptake of screening and late presentations [1, 13, 14].

South Africa has also implemented a population-based cervical cancer screening program that allows for at least three cervical smears within a woman's lifetime; performed at ten-year intervals and starting from the age of thirty [15, 16]. In this free program, the national screening coverage target is set at 70% of the eligible population [17]. However, studies suggest the suboptimal performance of this program in South Africa [4, 5, 9, 10, 18]. The District Health Barometers also report a decline in national coverage from 64.5% in 2016/17 to 61.2% in 2017/18, and Gauteng province, where the current study was conducted, had one of the lowest coverage at 47.7% [19].

Several factors have been reported in the literature to influence cervical cancer screening behaviours. Socio-demographic characteristics such as older age, formal education, and high economic status have been shown to influence cervical cancer screening uptake [20–22]. A study conducted in five Sub-Saharan African countries reported the highest cervical cancer screening uptake among women with the highest socioeconomic status, older age group (40–49 years), and secondary or higher educational attainment.

By implication, the lowest cervical cancer screening uptake was reported among women with the lowest socioeconomic status and younger age group (21–29 years) [20]. These findings were also confirmed in an Ethiopian study that found that high educational status and older age increased cervical cancer screening uptake by more than six and four-fold, respectively [23]. Studies have also suggested that women with high knowledge about cervical cancer are five times more likely to go for cervical cancer screening [3, 24]. Clinical factors have also been reported associated with cervical cancer screening. An Ethiopian study found that women who have been treated for sexually transmitted infections (STI) were five times more likely to be screened for cervical cancer compared to women who have not [23]. Also, infection with HIV predicted cervical cancer screening uptake since the integration of both services makes HIV clinics to act as additional sources of information for cervical cancer risks and a platform for screening [23]. However, a South African study among HIV-positive women found only a minority of HIV-positive women (28.6%) had optimal cervical cancer screening practices [25]. Overall, in South Africa, there are sociodemographic disparities in the burden of cervical cancer and screening practices, with black women reported having a higher burden, lower screening rates, poorer access, and more lag-time in presentation for treatment, and higher complications and mortality rates [26, 27]. Understanding cervical cancer screening

behaviours among black women in South Africa is therefore a public health imperative, particularly in the context of high prevalence of risk factors such as HIV.

The reasons for the low uptake of cervical screening vary depending on the setting and are important considerations for developing meaningful and context-appropriate interventions to increase screening uptake. In a study among ethnic minorities in the UK, key barriers to cervical screening included emotional barriers (fear, embarrassment, and shame), lack of time to come for the procedure, low perceived risk, and absence of symptoms [28]. However, in the Cape region of South Africa, insufficient information from primary care providers, negative community opinions relating to the procedure, and fear of having an HIV test at the same time, were prominent barriers cited by women for poor screening practices [29].

The recently proposed Integrated screening action model (I-SAM) (Fig 1) [30] critiques that screening behaviours are complex and existing models do not provide adequate frameworks for the explanation of these behaviours. The proponents of I-SAM posited that many models focus on how people who are already engaged with the healthcare system make decisions to screen or not to, but do not adequately explore the behaviours of those who are unaware or aware but not engaged. Also, screening for many diseases is a repetitive process and explanatory models should not focus only on how screening behaviours are initiated but also on how the behaviour is maintained. Furthermore, interventions cannot be one-size-fits-all since different persons are at different stages of the screening behaviour and require interventions targeted to their behavioural stages and circumstances.

Applied to cervical cancer screening, the I-SAM implicates 3 key tasks for healthcare providers: that not all women are at the same stage of the screening journey and screening engagements and interventions need to be informed by this; that the screening behavior of a woman is influenced by the circumstances, opportunities, and constraints within her environment; and that interventions need to focus on the source(s) of the behavior and create opportunities,

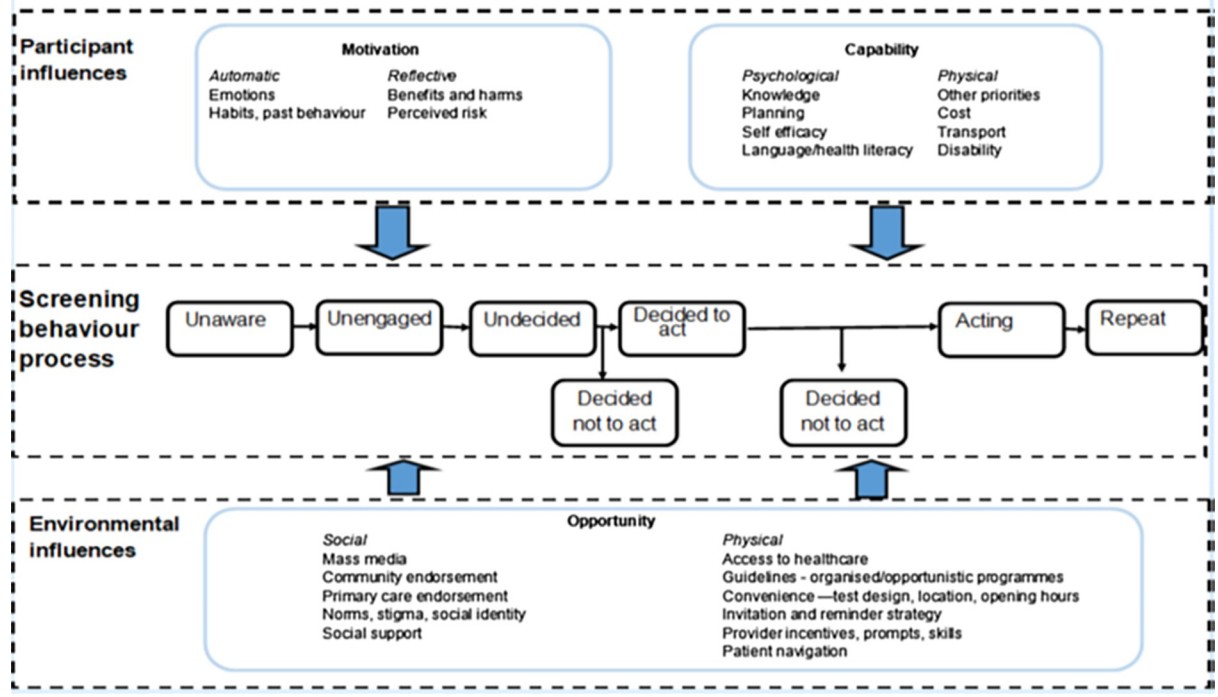

**Fig 1. The integrated screening action model.**

build capability and support the woman's intrinsic motivations to make an educated decision and take action by actually doing the cervical smear test.

Although several of the studies summarized above have illuminated on factors associated with screening uptake, they are limited in their explanations of the behavioural dynamics of study participants, particularly in the context of high prevalence of risk factors such as HIV. The aim of this study was to determine the uptake and factors associated with cervical cancer screening within the past 10-years among black women attending PHC facilities in Gauteng province, South Africa. In this article, we explain the study findings using the I-SAM and inform interventions that could increase the uptake of cervical cancer screening in South African PHC and similar settings.

## Methods

### Study design and setting

This was a sub-study of a Cross sectional study that aimed to determine an association between smokeless tobacco (SLT) use and cervical dysplasia. We recruited consecutive women who presented to the cervical screening programs of PHC clinics across the five health districts in Gauteng province–Johannesburg, Tshwane, Ekurhuleni, Sedibeng, and West Rand, between 2017 and 2020. The clinics were randomly selected from each district cluster. The number of clinics per district was proportional in respect to the proportion each district contributed to the total provincial Pap smear count for 2013/14 [31]. Considering the primary objective of the main study, we estimated a minimum sample of 1032; assuming a 95% confidence level, 80% power, 8.4% SLT use prevalence among women and a 6.5% incidence rate of abnormal Pap smears. We adjusted upwards by 10% for inadequate smears and/ or incomplete or lost data [32, 33], resulting in a sample size of 1135. Furthermore, to reduce the effects of intra and inter cluster correlations (ICC), we calculated the Effective sample size (ESS) using the equation: ESS = mk / [1+p(m-1)], where: 1+p(m-1) equals the design effect (DE), K equals the number of clinic cluster, ESS equals the effective sample size, m equals the number of participants per cluster / clinic and p equals the ICC coefficient [34]. Assuming an ICC coefficient (p) of 0.01 and an ESS of 1135, an m of 40 participants per clinic and a total of 40 clinics (k) across the 5 districts of Gauteng province were determined and a final sample size of 1600 – forty (40) to be recruited across Johannesburg (17 clinics), Tshwane (7 clinics), Ekurhuleni (11 clinics), Sedibeng (2 clinics), and West Rand (3 clinics). However, data collection was halted due to the COVID-19 pandemic restrictions and only 749 participants were recruited across 20 clinics in the five districts–Johannesburg (315), Ekurhuleni (170), Tshwane (79), West Rand (105) and Sedibeng (80).

### Recruitment of participants

We recruited consecutive women who presented to the cervical screening programs of PHC clinics across the five health districts in Gauteng province–Johannesburg, Tshwane, Ekurhuleni, Sedibeng, and West Rand, between March 2017- April 2020. Women who were eligible were approached at the vital signs station by a trained research assistant to participate in the study. The research assistants explained the nature and objective of the study to each woman and in addition, gave a participant information leaflet. Further explanation and clarifications were offered as needed. Those who indicated an interest in the study were directed to a private room where written informed consent was obtained, and the questionnaire administered. Thereafter, participants were taken to the nurse who then performed the cervical smear as per usual care or if not yet time, back to their positions on the service queue. Recruitment of participants continued until 40 participants were recruited in each clinic. Patients who were too ill

and pregnant were excluded from the study. Non-participation did not lead to disadvantages in service delivery. We did not include the information of those who did not consent to participate, which could have led to information bias the study was voluntary, hence those who did not consent to participate in the study were left in the queue.

## Tool and data collection

The measurement tool was a semi-structured, researcher-administered questionnaire developed de novo based on the literature [35–39]. The questionnaire was peer-reviewed and after ethics clearance piloted at one clinic that did not participate in the main study. Data from the pilot study was not included in the main study. The questionnaire collected information on participants'demographics (age, level of education, occupational status, and marital status), tobacco use (patterns of cigarette smoking, exposure to second-hand smoke, use of snuff and chewing of tobacco leaves), gynaecological information (menarche, parity, use of Family planning methods, use of hormone replacement therapy, coitarche, and use of barrier methods), social and behavioural information (HIV, number sexual partners, knowledge, attitudes, and cervical smear screening behaviours). Trained research assistants who were fluent in English and other local languages administered the questionnaire to the participants.

## Data analysis

The data was captured and analysed using Stata 16, 1 [40]. A total of 749 women were recruited for the primary objective. 77/749 (11%) were excluded from this analysis because this study focused on those women who self-identified as blacks. Black women were included because they are patrons of public health services in South Africa. Furthermore, other races were excluded from the study because their numbers were too small to provide a reliable estimate related to the objectives of the study. We used descriptive statistics such as frequencies to describe the study participants in terms of socio-demography, relevant sexual behaviour, HIV status, tobacco use, history of STI, knowledge of the risks of cervical cancer, and sources of information on cervical cancer and screening. The proportion that screened for cervical cancer in the last 10 years and the reasons for screening or not, were also determined using descriptive statistics. Associations between the cervical cancer screening uptake and sociodemographic characteristics, relevant sexual behaviour, HIV status, tobacco use, history of STI, knowledge of the risks of cervical cancer, and sources of information on cervical cancer and screening were explored using bivariate analysis. All significant exposure variables were included into the multivariable logistic regression model. We reported adjusted odds ratios, their corresponding 95% confidence intervals and p values. A p-value <0.05 was considered statistically significant.

## Ethical considerations

The study was approved by the Human Research Ethics Committee (HREC) of the University of Witwatersrand, Johannesburg Certificate number M160209. Participation in the study was voluntary and participants signed written informed consent. Confidentiality was ensured and data anonymized. Permission to conduct the study was obtained from the management of the participating districts.

## Results

Out of 749 of participants who were recruited, 672 were included in the analysis as they followed the inclusion criteria, 77/749 (11%) were excluded because they followed the exclusion

criteria. Majority of participants (81%) were above the age of 30 years, 75% attended at least high school, 64% unemployed. Most of the women (73%) had their early sexual debut during teenage period. Though most of the women were single (60%), seventy-five percent had one sex partner at the time of study while 88% have had two or more sex partners during their lifetime. Regardless of 57% of women receiving information about Pap smear from health facilities, only 35% had high knowledge on cervical cancer risks. The uptake of cervical cancer screening in the last 10 years was 42% (Table 1).

### Cervical cancer screening uptake among different socio-demographic factors

Women who were widowed, aged fifty years or more, having primary school level of education, without a sexual partner in the last 12 months, having low level of knowledge on cervical cancer risk factors and receiving information on cervical cancer from health facility platforms were factors associated with cervical cancer screening (Table 2).

### Factors associated with the uptake of cervical cancer screening in the past 10 years

Table 3 shows that women between 30–49 years old and 50 years or older, were five (CI: 2.53–8.73, p = <0.001) and ten (CI: 4.41–22.74, p = <0.001) times more likely to screen for cervical cancer when compared to 18–29-year-old women respectively. Women who were divorced (CI: 0.15–0.94, p = 0.04) and separated (CI: 0.10–0.86, p = 0.02) were less likely to report having screened for cervical cancer as compared to those who were single. Women with high knowledge on cervical cancer risk factors were less likely to report for screening (CI: 0.32–0.79, p = 0.003) as compared to those with low knowledge.

In Table 4, the most prevalent reason reported by participants for screening for cervical cancer was early first sexual experience (59.7%) followed screening as part of clinical management or periodic checkup.

In Table 5, the commonest reason for not screening for cervical cancer was lack of knowledge regarding Pap smear (35.4%).

## Discussion

Cervical cancer screening is a priority public health program and despite the implementation of a free and nationwide screening program in South Africa, only 42.4% of participants in this study reported screening in the last ten years. This low screening rate is like that reported in the 2013/14 Gauteng province report [31] and suggests that there has been no significant change in the provincial coverage in the last 8 years. However, better coverage of 63.2% was reported in another study conducted in PHC facilities in Johannesburg in 2017 [41], suggesting that there may be pockets of good performances within the same province and that targeted interventions that are informed by local health systems and socio-demographic factors in each district, are probably what is needed to improve the uptake of cervical cancer screening in Gauteng province.

As shown in Table 3, the proportion of participants that screened in the past 10-years varied depending on sociodemographic factors. The increased screening coverage in older age groups aligns with results of previous South African studies and elsewhere, and may be explained by the age eligibility criterion for cervical cancer screening [24]. Also, since the incidence of cervical cancer increases with age, older women are more likely to find screening beneficial and applicable to their health needs. However, to improve coverage, younger women especially

**Table 1. Socio-demographic characteristics of black women attending Pap smears in primary hea1lth care clinics in Gauteng Province, South Africa (N = 672).**

| Variable | Category | Total (N) = 672 | % |
|---|---|---|---|
| **Age** | 18–29 | 124 | 18.5 |
| | 30–49 | 416 | 61.9 |
| | >50 | 127 | 18.9 |
| | Missing | 5 | 0.7 |
| **Education status** | Did not attend school | 17 | 2.5 |
| | Primary school | 87 | 12.9 |
| | High school | 504 | 75.0 |
| | Post high school | 60 | 8.9 |
| | Missing | 4 | 0.6 |
| **Occupation status** | Unemployed | 430 | 64.0 |
| | Employed | 238 | 35.4 |
| | Missing | 4 | 0.6 |
| **Marital status** | Single | 403 | 59.9 |
| | Married | 157 | 23.4 |
| | Divorced | 28 | 4.2 |
| | Widowed | 41 | 6.1 |
| | Separated | 43 | 6.4 |
| **HIV status** | Negative | 310 | 46.1 |
| | Positive | 319 | 47.5 |
| | Missing | 43 | 6.4 |
| **Ever treated for STD** | No | 482 | 71.7 |
| | Yes | 179 | 26.6 |
| | Missing | 11 | 1.6 |
| **Age at first intercourse (years)** | 10–12 | 14 | 2.1 |
| | 13–19 | 489 | 72.8 |
| | 20–29 | 148 | 22.0 |
| | Missing | 21 | 3.1 |
| **Current sexual partner** | Yes | 515 | 76.6 |
| | Never | 0 | 0.0 |
| | Not now | 154 | 22.9 |
| | Missing | 3 | 0.4 |
| **Number of lifetime sex partners** | 1 or none | 65 | 9.7 |
| | 2 or more | 588 | 87.5 |
| | Missing | 19 | 2.8 |
| **Number of sex partners in last 12 months** | 0 | 111 | 16.5 |
| | 1 | 502 | 74.7 |
| | ≥2 | 47 | 7.0 |
| | Missing | 12 | 1.8 |
| **Cervical cancer screening in the last 10 years** | Yes | 284 | 42.3 |
| | No | 383 | 57.0 |
| | Missing | 5 | 0.7 |

those at significant risk but who may be <30-years-old and HIV-negative also, need to be targeted for screening, especially that a high prevalence of HIV and high-risk HPV infections have been reported in this population in South Africa [42].

The finding that separated and divorced participants were less likely to report screening compared to single women (Table 3). It is possible that in our study, single women perceived

**Table 2. Cervical cancer screening uptake according to various factors of the participants among black women in the past 10 years attending Pap smears in primary health care clinics in Gauteng Province, South Africa (N = 672).**

| Characteristic | Categories | No | | Yes | | P value |
|---|---|---|---|---|---|---|
| | | N | % | N | % | |
| **Age[1]** | 18–29 | 106 | 85.5 | 18 | 14.5 | <0.001 |
| | 30–49 | 227 | 54.6 | 189 | 45.4 | |
| | >50 | 50 | 39.4 | 77 | 60.6 | |
| **Education status[1]** | Did not attend school | 11 | 64.7 | 6 | 35.3 | 0.012 |
| | Primary school | 36 | 41.4 | 51 | 58.6 | |
| | High school | 302 | 59.9 | 202 | 40.1 | |
| | Post high school | 36 | 60.0 | 24 | 40.0 | |
| **Marital status** | Single | 231 | 57.3 | 172 | 42.7 | <0.001 |
| | Married | 84 | 53.5 | 73 | 46.5 | |
| | Divorced | 19 | 67.9 | 9 | 32.1 | |
| | Widowed | 16 | 39.0 | 25 | 61.0 | |
| | Separated | 37 | 86.0 | 6 | 14.0 | |
| **HIV status[1]** | Negative | 191 | 61.6 | 119 | 38.4 | 0.061 |
| | Positive | 173 | 54.2 | 146 | 45.8 | |
| **Number of sex partners in last 12 months[1]** | 0 | 52 | 46.8 | 59 | 53.2 | 0.037 |
| | 1 | 298 | 59.4 | 204 | 40.6 | |
| | ≥2 | 30 | 63.8 | 17 | 36.2 | |
| | Total | 380 | 57.6 | 280 | 42.4 | |
| **Ever treated for STD[1]** | No | 285 | 59.1 | 197 | 40.9 | 0.098 |
| | Yes | 93 | 52.0 | 86 | 48.0 | |
| | Total | 378 | 57.2 | 283 | 42.8 | |
| **Knowledge on cervical cancer risks[1]** | Low | 113 | 50.7 | 110 | 49.3 | <0.001 |
| | Medium | 112 | 55.2 | 91 | 44.8 | |
| | High | 159 | 68.2 | 74 | 31.8 | |
| **Sources of Pap smear information[1]** | Community platforms[3] | 165 | 65.0 | 89 | 35.0 | 0.006 |
| | Health facility platforms[4] | 207 | 54.0 | 176 | 46.0 | |
| | Missing (35) | | | | | |

[1]Varying totals because of missing values

**Table 3. Factors associated with cervical cancer screening in the last 10 years among black women in the past 10 years attending Pap smears in primary health care clinics in Gauteng Province, South Africa (N = 67).**

| Characteristic | Categories | Adjusted OR | 95% CI | p value |
|---|---|---|---|---|
| **Age (years)** | 18–29 | 1 | | |
| | 30–49 | 4.70 | 2.53–8.73 | **<0.001** |
| | Above 50 | 10.02 | 4.41–22.74 | **<0.001** |
| **Marital status** | Single | 1 | | |
| | Divorced | 0.37 | 0.15–0.94 | **0.036** |
| | Married | 0.84 | 0.53–1.33 | 0.454 |
| | Widowed | 1.39 | 0.59–3.26 | 0.438 |
| | Separated | 0.30 | 0.10–0.86 | **0.025** |
| **Knowledge on cervical cancer risks** | Low | 1 | | |
| | Medium | 0.80 | 0.51–1.26 | 0.345 |
| | High | 0.49 | 0.32–0.79 | **0.003** |

**Table 4. Reasons for taking the cervical cancer screening test among black women in the past 10 years attending Pap smears in primary health care clinics in Gauteng Province, South Africa (N = 672).**

| Reason | Responses | N | % |
| --- | --- | --- | --- |
| Early first sexual experience | No | 265 | 39.4 |
|  | Yes | 401 | 59.7 |
|  | Missing | 6 | 0.9 |
| Part of periodic health check | No | 498 | 74.8 |
|  | Yes | 168 | 25.0 |
|  | Missing | 6 | 0.9 |
| Part of clinical management | No | 493 | 73.4 |
|  | Yes | 173 | 25.7 |
|  | Missing | 6 | 0.9 |
| I know women affected by cervical cancer | No | 570 | 84.8 |
|  | Yes | 96 | 14.3 |
|  | Missing | 6 | 0.9 |
| To clean the womb | No | 531 | 79.0 |
|  | Yes | 135 | 20.1 |
|  | Missing | 6 | 0.9 |
| Vaginal bleeding during sex | No | 637 | 95.9 |
|  | Yes | 27 | 4.1 |
|  | Missing | 664 | 100.0 |
| Vaginal bleeding other times | No | 625 | 93.0 |
|  | Yes | 39 | 5.8 |
|  | Missing | 8 | 1.2 |
| Other personal concerns | No | 596 | 88.7 |
|  | Yes | 68 | 10.1 |
|  | Missing | 8 | 1.2 |

themselves as vulnerable to the risk factors for cervical cancer because of their less stable social relationships, and therefore willing to be screened. Married, separated, and divorced women may have a false sense of security that they are having or had a stable relationship and therefore screening is not necessary for them. This category are people who are not willing to screen and therefore decided not to act based on their perception of not being at risk of cervical cancer. Such a false sense of security raises serious concerns in this population with a high prevalence of HIV (known risk factors for cervical cancer) and compounded with the fact that male partners in South Africa do not always support their partners to screen for cervical cancer [43]. Thus, the targeted interventions to increase the uptake of cervical cancer screening in this group would be motivation and capability of the participants through environmental influence like mass media and social support.

Despite having national treatment guidelines stating that every HIV women should have a Pap smear at initiation of treatment, less than half (46%) of HIV positive women screened for cervical cancer [38, 40–42]. However, this contradicts Ugandan study, which reported cervical cancer screening of 10% among HIV positive women [24], This suggests that integration may not suffice on its own but should be part of a multi-faceted strategy that incorporates activities that provide opportunities for serial health education (to increase awareness of risks and the need to engage the screening services), personalize risks during the clinical encounter (to motivate non-deciders to take a decision), explore and support self-efficacy (to increase the chances of women acting on their decision to test.t) and increase access (to optimize screening uptake when a woman has decided to test).

**Table 5. Reasons for not taking the cervical cancer-screening test among black women in the past 10 years attending Pap smears in primary health care clinics in Gauteng Province, South Africa (N = 672).**

| Reason | Responses | N | % |
|---|---|---|---|
| **I do not know what is a pap smear** | No | 250 | 37.2 |
| | Yes | 238 | 35.4 |
| | Missing | 184 | 27.4 |
| **It is unnecessary for me** | No | 410 | 61.0 |
| | Yes | 78 | 11.6 |
| | Missing | 184 | 27.4 |
| **I did not know I had to do one** | No | 379 | 56.4 |
| | Yes | 109 | 16.2 |
| | Missing | 184 | 27.4 |
| **I have not been sexually active** | No | 482 | 71.7 |
| | Yes | 6 | 0.9 |
| | Missing | 184 | 27.4 |
| **Religious or cultural reasons** | No | 486 | 72.3 |
| | Yes | 2 | 0.3 |
| | Missing | 184 | 27.4 |
| **Bad attitudes of Doctors and nurses** | No | 483 | 71.9 |
| | Yes | 5 | 0.7 |
| | Missing | 184 | 27.4 |
| **I have not been sick to warrant it** | No | 473 | 70.4 |
| | Yes | 15 | 2.2 |
| | Missing | 184 | 27.4 |
| **I am afraid of positive test results** | No | 437 | 65.0 |
| | Yes | 51 | 7.6 |
| | Missing | 184 | 27.4 |
| **I could not access the clinic for the procedure** | No | 478 | 71.1 |
| | Yes | 10 | 1.5 |
| | Missing | 184 | 27.4 |

The knowledge of cervical cancer risk factors was negatively associated with screening, which contradicts the findings of previous studies [20, 23, 44]. It is also possible that women who were armed with information, conceivably, do not engage in risky sexual behaviors and hence, may not see a reason to screen. Healthcare professionals may need to employ more intense person-centered health education and counseling techniques (e.g. brief motivational interviewing) [45], to support women to make a behavioral shift from just being informed or aware of the need to screen, to deciding to do the cervical smear test, irrespective of their risk perception of their sexual activity. Providing such support may increase the number who test as patients often regard their healthcare providers as credible sources of information and positive influence [46]. This is more so that in this study, most participants obtained information on cervical cancer screening from their healthcare facilities. However, health education and counseling must be tailored to individual needs and circumstances since patients are not all at the same stage of behavioral change.

Twenty percent of participants had "clean my (their) womb" as a reason for cervical cancer screening and majority did not know what Pap smear was, and hence they did not take the cervical cancer screening in the past 10 years. This reflects significant gaps in patients' knowledge and highlights the need for healthcare professionals to routinely explore patients' basic knowledge and address the misconceptions of cervical cancer screening and the need for

reinforcement of the national guidelines to healthcare workers. This is a barrier to screening and reaffirms the need for more person-centered health education and intense counseling techniques. However, providing information only, may not be sufficient to make disinclined abstainers (people who are not inclined to screen and don't) and inclined abstainers (people who are inclined to screen but fail to act) to take up screening [30], particularly so, that women who had a high level of knowledge of risk factors in this study were significantly less likely to report screening. Integrating cervical cancer screening into all PHC services and creating prompts for clinicians during clinic visits by incorporating a question on the last time an eligible woman had a cervical smear within the vital signs, could increase the screening uptake.

Summed together, this study found low uptake of cervical cancer screening and the need for a more intensive and patient-centered health education and counseling program that translates women's knowledge into screening actions in South African PHC. Using the I-SAM (Fig 1), the low screening rate possibly hinges on participants' poor health literacy (poor knowledge and lack of awareness of the risk factors for cervical cancer), few opportunities of screening (since cervical cancer screening is a vertical program and poorly integrated into other PHC programs) and poor self-efficacy (poor translation of knowledge into screening behavior) [30]. Problems in any of these areas should then be addressed. Increasing community-based campaigns and use of social media platforms may further increase awareness and disseminate information on cervical cancer, the screening process and increase screening uptake [47]. Since the intrinsic drive of each patient is crucial to acting on decisions, healthcare professionals need to be upskilled in how to motivate patients to act on their decisions by exploring emotions, habits, past behaviors, perceived benefits, harms, and risks that influence their screening behaviors.

## Study limitations

Although the sample was large for a cross-sectional study, the relative under-representation of participants from some districts due to the need to stop recruitment because of the COVID-19 pandemic could limit the generalization of the study findings to all black women in Gauteng PHC facilities. We further acknowledge the potential for sampling bias when convenience sampling method is used. Since only black women's data were analyzed, the results may not be representative of the entire South African women population. However, black women by far are the patrons of public healthcare services in South Africa and are disproportionately affected by poor outcomes relating to cervical cancer, making it a public health imperative to focus on this subpopulation. The study was based on self-reports and therefore has potential for social desirability. This was addressed by the openness, transparency, and nonthreatening engagement of the trained field workers with participants. Since study participants were women who had come to the clinics for cervical cancer screening of their own volition, they were more likely to report better knowledge and screening behaviors than the larger clinic women cohort or those who had not visited the clinic. The true prevalence of screening in the past 10-years among black women in the general population would therefore be probably less than that reported in this study. Furthermore, caution needs to be exercised not to over-estimate the association between good knowledge and not screening in the last 10 years since participants with good knowledge at the time of this study but who had not screened in the past, could have had poor knowledge then, and therefore not screened in the past. The study was not able to account for how many women who were screened came for treatment according to the I-SAM model. Notwithstanding these limitations, this study provides insight into the past screening behaviors of black women in this urban PHC setting and indicates the need for targeted interventions and integrated PHC services to promote cervical cancer screening.

## Conclusion

Cervical cancer screening uptake in the past 10-years among black South African women attending PHC is low. The lack of awareness of risk factors and the necessity of screening for cervical cancer call for strategies to strengthen health education and prevention programs targeted to each patient's context and stage of behavioral change. Such strategies must consider the socio-demographic and clinical correlates of cervical cancer screening in South African PHC and promote its integration into other PHC services.

## Acknowledgments

We would like to thank all the women who participated in the study and the staff of PHC clinics in the districts that contributed to the success of this project.

## Author Contributions

**Conceptualization:** Joyce Sikwese Musonda, Olalekan Ayo-Yusuf, Elizabeth Reji, John Musonda, Langalibalele Honey Mabuza, John Velaphi Ndimande, Jimmy Akii, Olufemi Babatunde Omole.

**Data curation:** Joyce Sikwese Musonda, Elizabeth Reji, John Musonda, Langalibalele Honey Mabuza, John Velaphi Ndimande, Jimmy Akii, Olufemi Babatunde Omole.

**Formal analysis:** Pumla Pamella Sodo, Olalekan Ayo-Yusuf, Elizabeth Reji, John Musonda, Langalibalele Honey Mabuza, John Velaphi Ndimande, Jimmy Akii, Olufemi Babatunde Omole.

**Software:** Pumla Pamella Sodo.

**Validation:** Olufemi Babatunde Omole.

**Writing – original draft:** Joyce Sikwese Musonda, Pumla Pamella Sodo, Elizabeth Reji, John Musonda, Langalibalele Honey Mabuza, John Velaphi Ndimande, Jimmy Akii, Olufemi Babatunde Omole.

**Writing – review & editing:** Joyce Sikwese Musonda, Pumla Pamella Sodo, Olalekan Ayo-Yusuf, Olufemi Babatunde Omole.

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
