## [Decision Letter · Decision Letter 0]

13 Jul 2022

PGPH-D-22-00828

Cervical cancer screening in a population of black South African women with high HIV prevalence: a cross-sectional study

Dear Pumla Pamella Sodo,

Thank you for submitting your manuscript to PLOS Global Public Health. After careful consideration, we feel that it has merit but does not fully meet PLOS Global Public Health’s publication criteria as it currently stands. Therefore, we invite you to submit a revised version of the manuscript that addresses the points raised during the review process.

We look forward to receiving your revised manuscript.

Kind regards,

Nebiyu Dereje, MPH, PhD

Academic Editor

Journal Requirements:

1.  Please amend your detailed online Financial Disclosure statement. This is published with the article. It must therefore be completed in full sentences and contain the exact wording you wish to be published.

State what role the funders took in the study. If the funders had no role in your study, please state: “The funders had no role in study design, data collection and analysis, decision to publish, or preparation of the manuscript.”

2. Please update your online Competing Interests statement. If you have no competing interests to declare, please state: “The authors have declared that no competing interests exist.”

3. In the online submission form you indicate that your data is not available for proprietary reasons and have provided a contact point for accessing this data. Please note that your current contact point is a co-author on this manuscript. According to our Data Policy, the contact point must not be an author on the manuscript and must be a third party. Please revise your data statement to a non-author institutional point of contact, such as a data access or ethics committee, and send this to us via return email. Please also include contact information for the third party organization, and please include the full citation of where the data can be found.

4. Please provide separate figure files in .tif or .eps format and ensure that all files are under our size limit of 10MB.

5. Please change the item type of Figure 1 from 'Supporting Information' to 'Figure'.

6. Please include a legend for Figure 1 in your manuscript.

7. We have noticed that you have uploaded Supporting Information files, but you have not included a list of legends. Please add a full list of legends for your Supporting Information files after the references list.

Additional Editor Comments (if provided):

Reviewer 1:

This manuscript highlights the pragmatic reality of the sub-Saharan African situation in regard to the uptake of screening for cervical cancer. According to WHO, the Global strategy to accelerate the elimination of cervical cancer as a public health problem recommends that 70% of women are screened with high-performance tests by ages 35 and 45 years. As rightly pointed out in this paper, that many models focus on how people who are already engaged with the healthcare system make decisions to screen or not to, but do not adequately explore the behaviors of those who are unaware or aware but not engaged.

It made for very interesting reading. However below are a few of my comments for your consideration.

Introduction:

1. The study aim has clearly been articulated, apart from the inclusion criteria, “based on the high yield Gauteng report of 2013/2014”, the authors did not clearly specify the inclusion/exclusion criteria (e.g risk, stage of disease progression, women on chemotherapy already etc.) with sufficient detail and all necessary information critical to the study.

2. Line 128-129 the authors state that “The facilities were purposefully selected for their high yields in cervical cancer screening in the 2013/14 Gauteng clinics report”, what was the reason for not including low-moderate yield sites? Could they have been selectively excluded yet they could have been that way because of the insufficient information from primary care providers, negative community opinions relating to the procedure, and fear of having an HIV test at the same time? This needs to be articulated with clarity.

Methods:

1. In line 140-141, the authors state that 749 women were recruited, but only data for 672 were extracted for this study. What happened to the 77 who were excluded from the analyses? Please account for them.

2. Overall, there is no mention of the sample size. How did you justify your sample size? Please explain how the study size was arrived at.

3. The ‘recruitment and participants’ section are not clear. Was the selection process used likely to select participants that were representative of the population under investigation? Please clarify.

4. The authors should describe any efforts to address potential sources of bias.

5. What measures did the authors undertake to address and categorize non-responders? Please describe this.

6. Line 144-145 the authors state that “The measurement tool was a semi-structured, research-administered questionnaire developed de novo based on the literature (31-35)” how did the authors correctly measure the risk factors and outcome variables using a measurement tool that had not been trialed, piloted, or published previously? This is not clear please clarify.

7. Line 160-164 the authors mention using bivariate logistic regression models. However, there is no description of all the statistical methods used including those used to control for confounding. Can you include this too?

8. Describe any methods the authors used to examine subgroups and interactions.

9. The authors need to explain how missing data were addressed.

10. The authors should describe any sensitivity analyses done.

Results:

1. In Table 1, apart from the marital status variable, all the other variables do not each total to 672 as stated by the authors, and yet the percentages total 100. Please revisit your numbers and report the correct ones.

2. The authors should report numbers of individuals at each stage of the study e.g., numbers potentially eligible, examined for eligibility, confirmed eligible, included in the study.

3. They should also give reasons for non-participation at each stage.

4. Line 179-180 the authors state “only 42.4% of participants reported screening for cervical cancer in the last 180 10 years” drawn from Table 2. However, this is not clear in Table 2. Please clarify.

5. Line 183-184 is incomplete. Where did the rest obtain information on cervical cancer screening from?

Other minor comments:

• There are a few typos to review. e.g., line 56-58, line 83,

• Line 149, HIV status and use of antiretroviral drugs is not part of gynecological information.

• A Data Availability Statement.

Reviewer 3:

This is an interesting report on HPV screening in the last 10 years among women found in primary health care in South Africa. It reports a prevalence and associated factors.

Some issues:

1. Is the 40 women per health facility something computed? Please add further details on the original sample size calculation.

2. Please share the data collection tool in the appendix.

3. For data analysis:

- Please do not write STATA. It is Stata and please cite.

- Line 162. It is multivariable logistic regression. Or multiple logistic regression

- Line 161 it is stated that only statistically significant on bivariate pass to the multivariable. Looking at your results on the table 3 it doesn't seem to be this.

4. I see a lot of characteristics on thable 3 that are not on table 1 or 2. Why?

5. For the screening prevalence it would be good to report the 95% confidence interval (in the abstract as well as in the main manuscript). So I would recommend changing table 2:

- Remove the total for each variable. This is unnecessary.

- Righ now the sequence of the columns is "characteristics, N, % (for No), N, % (for Yes), N (for total) and p-value. I propose to remove the N for total. Make the percentages per column; remove the p-values; Add a column for prevalence of screening (and its confidence interval).

6. Table 3:

- Remove the N and %. That is on table 2.

- For marital status please use te single or married as reference.

- What 1< means on the number of lifetime sex partners

7. Table 4:

- Please remove the subtotals per variable. Just add an overall row stating that the total is 666.

8. Table 5:

- The same comments as for table 4. The overall total here is 488.

Reviewer 4:

The study under review reports findings from a cross-sectional study in Gauteng, assessing uptake of screening for cervical pre-cancer in a high-risk population. Overall, the study is well conducted and reported. The following issues should be addressed:

• Attending screening is complex, because it is a process rather than a single event. For example, after the screening procedure women have to receive the test results and if positive return for treatment, before they return for follow-up and re-screening. In reality, several of these steps may fail and although woman may attend the screening procedure, the overall outcome may actually fail, if the woman e.g. did not undergo treatment of the precancerous lesion. In the I-SAM model this level of complexity is not well addressed (to my understanding) and the authors may want to include this in their discussion. In this regard, the WHO Elimination Strategy targets to not only screen 70% of women but also to treat 90% of those with pre-cancer. The authors should consider to make reference to the WHO CC Elimination Strategy.

• Please follow all steps of the STROBE guideline for reporting, including a report on the number of participants potentially eligible, screened for eligibility, finally included, as well as missing data for the different variables in the table in there are any. From the STROBE checklist

o a) Report numbers of individuals at each stage of study—eg numbers potentially eligible, examined for eligibility, confirmed eligible, included in the study, completing follow-up, and analysed

o (b) Give reasons for non-participation at each stage

o (c) Consider use of a flow diagram

o (b) Indicate number of participants with missing data for each variable of interest

• Please add total number of participants to table 1.

• Please consider to move Table 2 to the appendix.

• Please revise references in the text, the references should be set in front of the punctuation, not behind.

Reviewers' comments:

Reviewer's Responses to Questions

**Comments to the Author**

1. Does this manuscript meet PLOS Global Public Health’s publication criteria? Is the manuscript technically sound, and do the data support the conclusions? The manuscript must describe methodologically and ethically rigorous research with conclusions that are appropriately drawn based on the data presented.

Reviewer #1: Partly

Reviewer #3: Yes

Reviewer #4: Yes

2. Has the statistical analysis been performed appropriately and rigorously?

Reviewer #1: I don't know

Reviewer #3: Yes

Reviewer #4: Yes

3. Have the authors made all data underlying the findings in their manuscript fully available (please refer to the Data Availability Statement at the start of the manuscript PDF file)?

Reviewer #1: No

Reviewer #3: No

Reviewer #4: Yes

4. Is the manuscript presented in an intelligible fashion and written in standard English?

Reviewer #1: Yes

Reviewer #3: Yes

Reviewer #4: Yes

5. Review Comments to the Author

Reviewer #1: The manuscript requires some major revisions in line with the STROBE Statement -Checklist of items that should be included in reports of cross-sectional studies. the results section has many tables which seem to be repetitive. consider where possible combining some of these tables.

Reviewer #3: This is an interesting report on HPV screening in the last 10 years among women found in primary health care in South Africa. It reports a prevalence and associated factors.

Some issues:

1. Is the 40 women per health facility something computed? Please add further details on the original sample size calculation.

2. Please share the data collection tool in the appendix.

3. For data analysis:

- Please do not write STATA. It is Stata and please cite.

- Line 162. It is multivariable logistic regression. Or multiple logistic regression

- Line 161 it is stated that only statistically significant on bivariate pass to the multivariable. Looking at your results on the table 3 it doesn't seem to be this.

4. I see a lot of characteristics on thable 3 that are not on table 1 or 2. Why?

5. For the screening prevalence it would be good to report the 95% confidence interval (in the abstract as well as in the main manuscript). So I would recommend changing table 2:

- Remove the total for each variable. This is unnecessary.

- Righ now the sequence of the columns is "characteristics, N, % (for No), N, % (for Yes), N (for total) and p-value. I propose to remove the N for total. Make the percentages per column; remove the p-values; Add a column for prevalence of screening (and its confidence interval).

6. Table 3:

- Remove the N and %. That is on table 2.

- For marital status please use te single or married as reference.

- What 1< means on the number of lifetime sex partners

7. Table 4:

- Please remove the subtotals per variable. Just add an overall row stating that the total is 666.

8. Table 5:

- The same comments as for table 4. The overall total here is 488.

Reviewer #4: The study under review reports findings from a cross-sectional study in Gauteng, assessing uptake of screening for cervical pre-cancer in a high-risk population. Overall, the study is well conducted and reported. The following issues should be addressed:

• Attending screening is complex, because it is a process rather than a single event. For example, after the screening procedure women have to receive the test results and if positive return for treatment, before they return for follow-up and re-screening. In reality, several of these steps may fail and although woman may attend the screening procedure, the overall outcome may actually fail, if the woman e.g. did not undergo treatment of the precancerous lesion. In the I-SAM model this level of complexity is not well addressed (to my understanding) and the authors may want to include this in their discussion. In this regard, the WHO Elimination Strategy targets to not only screen 70% of women but also to treat 90% of those with pre-cancer. The authors should consider to make reference to the WHO CC Elimination Strategy.

• Please follow all steps of the STROBE guideline for reporting, including a report on the number of participants potentially eligible, screened for eligibility, finally included, as well as missing data for the different variables in the table in there are any. From the STROBE checklist

o a) Report numbers of individuals at each stage of study—eg numbers potentially eligible, examined for eligibility, confirmed eligible, included in the study, completing follow-up, and analysed

o (b) Give reasons for non-participation at each stage

o (c) Consider use of a flow diagram

o (b) Indicate number of participants with missing data for each variable of interest

• Please add total number of participants to table 1.

• Please consider to move Table 2 to the appendix.

• Please revise references in the text, the references should be set in front of the punctuation, not behind.

6. PLOS authors have the option to publish the peer review history of their article (what does this mean?). If published, this will include your full peer review and any attached files.

**Do you want your identity to be public for this peer review?** For information about this choice, including consent withdrawal, please see our Privacy Policy.

---

## [Decision Letter · Decision Letter 1]

10 Oct 2022

PGPH-D-22-00828R1

Cervical cancer screening in a population of black South African women with high HIV prevalence: a cross-sectional study

Dear Dr. Sodo,

Thank you for submitting your manuscript to PLOS Global Public Health. After careful consideration, we feel that it has merit but does not fully meet PLOS Global Public Health’s publication criteria as it currently stands. Therefore, we invite you to submit a revised version of the manuscript that addresses the points raised during the review process.

We look forward to receiving your revised manuscript.

Kind regards,

Nebiyu Dereje, MPH, PhD

Academic Editor

Journal Requirements:

Additional Editor Comments (if provided):

Please, kindly address the following comments from the reviewers.

Reviewer 1:

Introduction

Line 149, include the month of the year in your statement “between 2017 and 2020”

Results

1. In Table 1, your percentages on “cervical cancer screening in the last 10 years does not add up.

Those who said yes, no and missing total 672, while the individual percentages do not total 100 % (42.3 + 57+ 0.07) what happened to 0.63 %?

2. Table 1 percentages are wrong and misleading, some exceed a sum of 100% or are far less than 100% for example education status, marital status, ever treated for STD, current sexual partner, number of lifetime sexual partners etc. please check your figures again and update them accordingly.

3. Line 166 -172; Using the de novo questionnaire tool, to capture information on “The questionnaire collected information on 167 participants’ demographics (age, level of education, occupational status, and marital status), 168 tobacco use (patterns of cigarette smoking, exposure to second-hand smoke…….” Can the authors explain why in table 2 they state N=672, and go on to put a disclaimer in line 467 that the totals varied because of missing values? Was this tool validated correctly?

4. Please let the authors review the figures in the other tables to assure that each value is accounted for both in absolute numbers and percentages.

This paper is well written albeit for these small numerical errors. In my opinion I do recommend this paper for publication after making the small revisions.

Reviewer 2:

This is the second revision of this interesting manuscript. My questions have been addressed. However, a few small things remain:

1. Please see the official documentation of Stata. It is a very common mistake to write STATA, as if it was an acronym. Please correct to be Stata.

2. Ok, the adjusted regression results. But you do not have to keep only bivariate significant factors. As long as you justify in the methods that, for example, you kept HIV regardless of the p-value.

3. The second paragraph of the introduction requires some clarification:

- Citation 6 - please find an additional reference to this. And please provide an URL (link) for citation 6.

- Lines 54 and 55, in the among the best possible protection against cervical cancer. Why the vaccine is not included here?

Reviewers' comments:

Reviewer's Responses to Questions

**Comments to the Author**

1. If the authors have adequately addressed your comments raised in a previous round of review and you feel that this manuscript is now acceptable for publication, you may indicate that here to bypass the “Comments to the Author” section, enter your conflict of interest statement in the “Confidential to Editor” section, and submit your "Accept" recommendation.

Reviewer #1: All comments have been addressed

Reviewer #3: All comments have been addressed

2. Does this manuscript meet PLOS Global Public Health’s publication criteria? Is the manuscript technically sound, and do the data support the conclusions? The manuscript must describe methodologically and ethically rigorous research with conclusions that are appropriately drawn based on the data presented.

Reviewer #1: Yes

Reviewer #3: Yes

3. Has the statistical analysis been performed appropriately and rigorously?

Reviewer #1: I don't know

Reviewer #3: Yes

4. Have the authors made all data underlying the findings in their manuscript fully available (please refer to the Data Availability Statement at the start of the manuscript PDF file)?

Reviewer #1: Yes

Reviewer #3: Yes

5. Is the manuscript presented in an intelligible fashion and written in standard English?

Reviewer #1: Yes

Reviewer #3: Yes

6. Review Comments to the Author

Reviewer #1: Introduction

Line 149, include the month of the year in your statement “between 2017 and 2020”

Results

1. In Table 1, your percentages on “cervical cancer screening in the last 10 years does not add up.

Those who said yes, no and missing total 672, while the individual percentages do not total 100 % (42.3 + 57+ 0.07) what happened to 0.63 %?

2. Table 1 percentages are wrong and misleading, some exceed a sum of 100% or are far less than 100% for example education status, marital status, ever treated for STD, current sexual partner, number of lifetime sexual partners etc. please check your figures again and update them accordingly.

3. Line 166 -172; Using the de novo questionnaire tool, to capture information on “The questionnaire collected information on 167 participants’ demographics (age, level of education, occupational status, and marital status), 168 tobacco use (patterns of cigarette smoking, exposure to second-hand smoke…….” Can the authors explain why in table 2 they state N=672, and go on to put a disclaimer in line 467 that the totals varied because of missing values? Was this tool validated correctly?

4. Please let the authors review the figures in the other tables to assure that each value is accounted for both in absolute numbers and percentages.

This paper is well written albeit for these small numerical errors. In my opinion I do recommend this paper for publication after making the small revisions.

Reviewer #3: This is the second revision of this interesting manuscript. My questions have been addressed. However, a few small things remain:

1. Please see the official documentation of Stata. It is a very common mistake to write STATA, as if it was an acronym. Please correct to be Stata.

2. Ok, the adjusted regression results. But you do not have to keep only bivariate significant factors. As long as you justify in the methods that, for example, you kept HIV regardless of the p-value.

3. The second paragraph of the introduction requires some clarification:

- Citation 6 - please find an additional reference to this. And please provide an URL (link) for citation 6.

- Lines 54 and 55, in the among the best possible protection against cervical cancer. Why the vaccine is not included here?

7. PLOS authors have the option to publish the peer review history of their article (what does this mean?). If published, this will include your full peer review and any attached files.

**Do you want your identity to be public for this peer review?** For information about this choice, including consent withdrawal, please see our Privacy Policy.

Reviewer #1: **Yes: **Dr Ivan Namakoola

Reviewer #3: **Yes: **Orvalho Augusto

---

## [Editor Report · Decision Letter 2]

13 Oct 2022

Cervical cancer screening in a population of black South African women with high HIV prevalence: a cross-sectional study

PGPH-D-22-00828R2

Dear Miss Sodo,

We are pleased to inform you that your manuscript 'Cervical cancer screening in a population of black South African women with high HIV prevalence: a cross-sectional study' has been provisionally accepted for publication in PLOS Global Public Health.

Best regards,

Nebiyu Dereje, MPH, PhD

Academic Editor